Efficacy and safety of low molecular weight heparin compared to unfractionated heparin for chronic outpatient hemodialysis in end stage renal disease: systematic review and meta-analysis

Palamaner Subash Shantha Ghanshyam 1 2 shanthag@thewrightcenter.org
Kumar Anita Ashok 1 2
Sethi Mansha 3
Khanna Rohit C. 4
Pancholy Samir Bipin 5
1 The Wright Center for Graduate Medical Education , Scranton, PA , USA
2 The Johns Hopkins University, Bloomberg School of Public Health , Baltimore, MD , USA
3 Temple University School of Medicine , Philadelphia, PA , USA
4 Allen Foster Research Centre for Community Eye Health, International Centre for Advancement of Rural Eye Care, L V Prasad Eye Institute , Hyderabad , India
5 Department of Medicine, The Commonwealth Medical College , Scranton, PA , USA
Tulkens Paul
Electronic publication date: 2015 Mar 10
Publication date: 2015
Volume: 3
Electronic Location ID: e835
Received 2014 Nov 28; Accepted 2015 Feb 22
Copyright: © 2015 Palamaner Subash Shantha et al.
Copyright year: 2015
Copyright holder: Palamaner Subash Shantha et al.
License: This is an open access article distributed under the terms of the Creative Commons Attribution License, which permits unrestricted use, distribution, reproduction and adaptation in any medium and for any purpose provided that it is properly attributed. For attribution, the original author(s), title, publication source (PeerJ) and either DOI or URL of the article must be cited.
License URL: https://creativecommons.org/licenses/by/4.0/

Keywords: Heparin, Hemodialysis, Thromboprophylaxis, Meta-analysis

Funding: The authors declare there was no funding for this work.

==============================
Background. Low molecular weight heparin (LMWH) is an effective anti-coagulant for thrombotic events. However, due to its predominant renal clearance, there are concerns that it might be associated with increased bleeding in patients with renal disease.

Objectives. We systematically evaluated the efficacy and safety of LMWH compared to unfractionated heparin (UH) in end stage renal disease (ESRD) patients.

Search Methods. Pubmed, Embase and cochrane central were searched for eligible citations.

Selection Criteria. Randomized controlled trials, comparing LMWH and UH, involving adult (age > 18 years), ESRD patients receiving outpatient, chronic, intermittent hemodialysis were included.

Data Collection and Analysis. Two independent reviewers performed independent data abstraction. I2 statistic was used to assess heterogeneity. Random effects model was used for meta-analysis.

Results. Nineteen studies were included for systematic review and 4 were included for meta-analysis. There were no significant differences between LMWH and UFH for extracorporeal circuit thrombosis [risk ratio: 1 (95% CI [0.62–1.62])] and bleeding complications [risk ratio: 1.16 (95% CI [0.62–2.15])].

Conclusions. LMWH is as safe and effective as UFH. Considering the poor quality of studies included for the review, larger well conducted RCTs are required before conclusions can be drawn.

Introduction

Chronic kidney disease (CKD) was prevalent in 25.8 million adults in the United States in 2004 (Snyder, Foley & Collins, 2009). CKD prevalence will increase by 5 million every decade in the United States (Rao et al., 2008). This alarming increase in CKD prevalence is due to an associated increase in the prevalence of hypertension, type 2 diabetes mellitus and obesity in the United States (Rao et al., 2008; Flegal et al., 2010; Finkelstein, Fiebelkorn & Wang, 2003). CKD, obesity, hypertension and diabetes in unison are estimated to cost the American health care system a sum of $110 billion annually (Mokdad et al., 2003; KDOQI; National Kidney Foundation, 2006).

Heparin acts by accelerating the inhibition of thrombin, and also factors (F) Xa, IXa, XIa and XIIa. Low molecular weight heparins (LMWH) are recently identified, widely used, heparin derivatives with a mean molecular weight of less than 8,000 Daltons (Linhardt & Gunay, 1999). Commonly used LMWH are Bemiparin, Certoparin, Dalteparin, Enoxaparin, Nadroparin, Parnaparin, Reviparin and Tinzaparin (Gould et al., 1999). They have a lower incidence of heparin induced thrombocytopenia (Gould et al., 1999; Gray, Mulloy & Barrowcliffe, 2008; Nicolaides, 2006) compared to UH (Gould et al., 1999). LMWHs, due to the shorter polysaccharide chain show a more pronounced FXa inhibitory profile, have a longer half-life and aids in once-a-day administration. However, action of LMWH depends on the length of the polysaccharide chain and hence, all LMWHs do not show the same inhibitory profile.

Observational studies showed that use of LMWH for prevention of extra-corporeal circuit thrombosis during dialysis sessions and graft or fistula thrombosis post dialysis in end stage renal disease (ESRD) patients were associated with greater bleeding risk compared to UH (European Pharmacopedia Commission, 1991; Gerlach, Pickworth & Seth, 2000). RCTs that assessed efficacy of LMWH had either excluded patients with renal disease or through inadequately powered sub-group analysis, had shown that patients with renal disease may be at risk for increased bleeding events (Spinler et al., 2003). A systematic review and meta-analysis on the same topic was conducted by Lim, Cook & Crowther (2004) in 2004 where they had abstracted data from 17 trials. They concluded that LMWH was as effective and safe as UH in patients with ESRD receiving regular hemodialysis (Lim, Cook & Crowther, 2004). However, as the authors had reported, risk of bias was high for the studies included in their meta-analysis and they were small population studies.

The rationale for our systematic review and meta-analysis are: (1) we have focused our comparison to LMWH and UH only. Our review will be clinically useful since 95% centers around the globe use only these 2 drugs and not citrate as analyzed in the review by Lim, Cook & Crowther (2004), (2) we have focused our review to only those LMWH that are currently approved by the Food and Drug Administration (FDA). Hence, our review will be clinically relevant for US dialysis centers, (3) we have only included studies that had an explicit random allocation. We excluded controlled clinical trials that did not have an explicit random allocation; the review by Lim, Cook & Crowther (2004) had included controlled trials that did not have an explicit random allocation. Hence, it is likely that our estimates are less biased. Hence via this systematic review and meta-analysis we have compared the efficacy and safety of LMWH compared to UH in patients with ESRD receiving outpatient, chronic, intermittent hemodialysis for dialysis associated events (examples: extra-corporeal circuit thrombosis graft/fistula thrombosis and bleeding complications). This review does not attempt to compare the 2 types of heparins for treatment of deep vein thrombosis or pulmonary embolism in these patients.

Methods

The protocol of this study can be found as Data S1. This protocol was approved by the Johns Hopkins Centers for Clinical Trials, Baltimore, Maryland, USA.

Data sources

We searched 3 databases namely: (1) Pubmed, (2) Embase, (3) Cochrane central. We did not use language or date restrictions when we searched for citations. Detailed search strategy has been explained in the Appendix. A librarian from Johns Hopkins University Bloomberg School of Public Health, Baltimore, Maryland, USA helped us develop the search strategy. Date of last search was 24th November 2014.

Study selection

Criteria for study inclusion in this review were: (1) RCTs comparing LMWH and UF; we only included studies where the intervention allocation was truly random. Quasi-randomized or any other types of non-random intervention allocation were criteria for exclusion. (2) We included any RCT that used LMWH approved by the FDA; this included Dalteparin, Enoxaparin, and Tinzaparin. (3) The study participants in the included studies were adult patients (age > 18 years) with end stage renal disease (ESRD), (4) The study participants were receiving chronic, intermittent, out-patient hemodialysis for renal replacement therapy. Only human studies were included. We excluded studies where LMWH was administered to patients not for the indication of anti-coagulation for hemodialysis but for therapy of another condition such as deep vein thrombosis, pulmonary embolism etc. Three reviewers independently assessed studies for eligibility. After title and abstract review, full texts of those citations determined eligible were assessed. From these full text citations, those that satisfied all criteria for inclusion were included in the review. Discrepancy was resolved by consensus.

Outcomes of our review

We focused on clinically relevant outcomes as primary outcomes for this review. They were: (1) extracorporeal circuit thrombosis during dialysis session: abstracted as presence or absence (yes/no), (2) graft or fistula thrombosis 7 days after trial drug administration (abstracted as yes/no); rationale being that we expected to remove confounding due to other factors that might play a role in graft and fistula thrombosis and hence 7 days would have been adequate time for the same. Other secondary outcomes considered are: (1) bleeding complications (i.e., intra-cranial hemorrhage, hemorrhagic stroke or any clinically recorded bleeding)—abstracted as number of patients with events; (2) deep vein thrombosis (DVT)—abstracted as number of patients with events; (3) pulmonary embolism (PE) (abstracted as number of patients with events); (4) vascular compression time (abstracted as continuous variable in seconds); (5) lipid profile: (low density lipoprotein (LDL), high density lipoprotein (HDL), very low density lipoprotein (VLDL), total cholesterol, LDL/HDL ratio))—(abstracted as continuous variables).

Other data abstracted from included studies

Type of RCT (including: year(s) of conduct, total sample size, study duration, date study commenced, place or region of study), study methodology (including eligibility criteria, methods of randomization, type of randomization sequence followed, allocation sequence concealment, and masking, washout period), participant characteristics: total number, setting (hospital based or free-standing), age, sex, country, race, comorbidities (diabetes, hypertension, bleeding disorders, autoimmune disorders), frequency of dialysis, intervention: LMWH and UH (dose, name of drug, route of drug, timing relative to hemodialysis, frequency of administration).

Assessment of risk of bias in included studies

Risk of bias was assessed by two independent reviewers. When there was a discrepancy, it was resolved by consensus. The studies were evaluated for the following criteria: (1) allocation, (2) masking of investigators and participants, (3) masking of outcome assessment, (4) loss to follow-up (attrition) and intention to treat analysis. Details of risk of bias assessment are mentioned in the protocol in the Supplemental Information.

Statistical analysis

We followed the analysis plan outlined in the protocol. We reported relative risks and 95% confidence intervals of the relative risks for all dichotomous outcomes. For continuous outcomes we calculated means, mean difference and standard deviations. Clinical heterogeneity was determined based on clinical knowledge. Methodological heterogeneity was assessed based on run in period, duration of study, adequacy of randomization etc. Statistical heterogeneity was assessed using I2. I2 of 25%, 50% and 75% was considered low, moderate and high heterogeneity respectively. If there were more than 10 included studies in the meta-analysis we had apriori decided to use funnel plots to assess reporting bias. Since our meta-analysis included only 4 studies we did not perform a funnel plot. Studies were pooled with the random effects model as we suspected significant heterogeneity; clinical, and methodological in the studies to be included in the meta-analysis. P < 0.05 was considered statistically significant. Although we had previously planned to do a sub-group analysis based on different types of LMWHs, we were unable to conduct sub-group analysis as planned due to the small number of included studies for meta-analysis. Although we had previously planed to perform sensitivity analysis based on study quality, since all included studies were similarly of poor quality we could not perform this sensitivity analysis.

Results

In total we had 4,095 citations retrieved after searching the three databases. After removing duplicates we were left with a final list of 3,735 citations. Of these, 19 citations were included in the review and 4 were pooled in the meta-analysis. Please refer to Fig. 1 for the study flow details.

Figure 1 Article flow diagram.

Details the process of study inclusion into the review

Characteristics of included studies (Table 1)

Among the 19 included RCTs (Aggarwal et al., 2004; Borm et al., 1986; Elisaf et al., 1997; Gritters et al., 2006; Harenberg et al., 1995; Hottelart et al., 1998; Lane et al., 1986; Lord et al., 2002; Mahmood et al., 2010; Naumnik, Borawski & Mysliwiec, 2003; Naumnik et al., 2007; Naumnik, Pawlak & Mysliwiec, 2007; Naumnik, Pawlak & Mysliwiec, 2009b; Naumnik, Pawlak & Mysliwiec, 2009a; Poyrazoglu et al., 2006; Ryan et al., 1990; Saltissi et al., 1999; Schrader et al., 1988; Verzan et al., 2004), 6 had a parallel group design and 13 had a cross-over design. Most regions of the world were represented, with 5 studies from Poland, 2 studies each from Netherland, Germany and United Kingdom and 1 study each from United States, Greece, France, Canada, Sweden, Turkey, Australia and Romania. Seven studies had evaluated enoxaparin, 6 dalteparin, and 5 evaluated tinzaparin. Sample size ranged from 8 to 70 participants. None of the studies explicitly defined a run-in period. Thirteen of the 19 studies had used a fixed LMWH dose and the remaining used a variable dose of LMWH. The fixed dose was usually between 0.69 mg/kg–1 mg/kg. For UH, most studies had used a bolus dose initially and was followed by an infusion. Thirteen studies had used a fixed dose of UH and 6 used variable dose of UH. The fixed dose comprised of 1,500–5,000 IU bolus followed by 36–62 IU/kg infusion.

Table 1 Characteristics of included studies.

Table details the characteristics of the studies included in the review.

Name of first author	Aggarwal	Borm	Elisaf	Gritters	Harenberg	Hottelart	Lane	Lord	Verzan	Naumnik	
Year	2004	1986	1997	2006	1995	1998	1986	2002	2004	2009	
Methods	Parallel RCT	Crossover RCT	Crossover RCT	Crossover RCT	Parallel RCT	Crossover RCT	Crossover RCT	Crossover RCT	Crossover RCT	Parallel RCT	
Participants	20	10	36	8	20	11	8	32	66	22	
Interventions	Enoxaparin	Dalteparin	Tinzaparin	Dalteparin	Dalteparin	LMWH not specified	Dalteparin	Tinzaparin	Tinzaparin	Enoxaparin	
Dose (fixed/variable)	Fixed	Fixed	Variable	Fixed	Fixed	Variable	Fixed	Variable	Variable	Fixed	
Country	United States	Netherlands	Greece	Netherlands	Germany	France	United Kingdom	Canada	Romania	Poland	
Notes							Difficult to assess randomization groups for outcomes		Abstract-specific data not provided for circuit thrombosis		
Outcomes	ADP-induced fibrinogen binding, platelet reactivity	Extracorporeal circuit thrombosis, bleeding complications, factor Xa levels, platelet function, beta-thromboglobulin, thromboxane A2, platelet factor 4, serotonin	Lipid profile (HDL, LDL, total cholesterol, apo A1, apo B, triglycerides, lipoprotein a), albumin, hemoglobin	Platelet factor 4, polymorphonuclear cells and platelet degranulation	Hep test, aPTT, thrombin clotting time	Plasma aldosterone, renin, aldosterone/renin ratio, serum potassium	Fibrinopeptide A, beta-thromboglobulin, kaolin cephalin clotting time, plasma heparin levels, bleeding time	Extracorporeal circuit thrombosis, bleeding complications, vascular compression time, patient/nurse satisfaction, relative cost, nursing time	Extracorporeal circuit thrombosis	Thrombomodulin, von-Willebrand factor, plasminogen activator inhibitor 1, cell surface adhesion molecule, e-selectin, intercellular adhesion molecule 1, prothrombin fragment 1 + 2	
Name of first Author	Mahmood	Naumnik	Naumnik	Naumnik	Naumnik	Poyrazoglu	Ryan	Saltissi	Schrader	
Year	2010	2003	2007	2007	2009	2006	1990	1999	1988	
Methods	Crossover RCT	Parallel RCT	Crossover RCT	Crossover RCT	Crossover RCT	Parallel RCT	Crossover RCT	Crossover RCT	Parallel RCT	
Participants	20	25	22	22	22	33	8	36	70	
Interventions	Tinzaparin	Enoxaparin	Enoxaparin	Enoxaparin	Enoxaparin	Dalteparin	Tinzaparin	Enoxaparin	Dalteparin	
Dose (fixed/variable)	Fixed	Fixed	Fixed	Fixed	Fixed	Fixed	Fixed	Variable	Variable	
Country	Sweden	Poland	Poland	Poland	Poland	Turkey	United Kingdom	Australia	Germany	
Notes							Abstract			
Outcomes	Lipid profile (LDL, HDL, total cholesterol, triglycerides) and lipoprotein lipase	Pro-thrombotic tissue factor, tissue factor pathway inhibitor, activated coagulation marker prothrombin fragment 1+2	Transforming growth factor beta-1, platelet derived growth factor AB, beta-thromboglobulin, platelet factor 4	Vascular endothelial growth factor, basic fibroblast growth factor	Prothrombin fragment1+2, thrombin/anti-thrombin complex	C-reactive protein, tumor necrosis factor alpha, superoxide dismutase, malondialdehyde	Anti-factor Xa activity, fibrinopeptide A, beta-thromboglobulin, hep test	Extracorporeal circuit thrombosis, bleeding complications, vascular compression time, lipid profile (LDL, HDL, VLDL, triglycerides, cholesterol)	Extracorporeal circuit thrombosis, bleeding complications, lipid profile, factor Xa level, erythrocyte concentration	

Patient characteristics (Table 2)

All study patients were adults (age > 18 years) with ESRD; ages ranged between 27–43 years. None of the study participants in any of the included studies had hyper-coagulable conditions or were receiving anti-coagulant or anti-platelet drugs. Diabetic nephropathy was the commonest in both these studies (33–36%) followed by hypertensive nephropathy (27%) and then by lupus nephritis (19%). All received regular, maintenance, outpatient hemodialysis. The dialysis frequency ranged between 3–5 sessions every week. Duration of dialysis was 4–5 h. Loss to follow-up ranged between 10–17% in the reported studies. Details of patient characteristics can be seen in Tables 1 and 2. Comorbid conditions of study participants are detailed in Table S1. The LMWH group and the UH group were similar with regards to the comorbid variables: diabetes prevalence, hypertension prevalence, coronary artery disease prevalence, smoking prevalence, obesity prevalence and dyslipidemia prevalence in most of the included studies (Table S1).

Table 2 Patient characteristics (patients in studies that reported outcomes of interest).

Characteristics of patients in the included studies.

Study, year (reference)	Mean age (yrs) ± SD	Excluded patients	Follow-up duration	Patients lost to follow-up	Patients (n): LMWH/UH	Frequency and duration of dialysis	Type of LMWH	Mean LMWH dose	Mean UFH dose	
		Other anticoagulants	Previous bleeding								
Borm et al., 1986	58.6	No	NS	NS	0/0	10	2-3/wk, 4 h	Dalteparin	(B) 18 IU/kg;
(I) 9 IU/kg/h	(B) 36 IU/kg;
(I) 18 IU/kg/h	
Saltissi et al., 1999	68.5	No	No	24 weeks	5	36	3–4/wk, 3–5 h	Enoxaparin	(B) 1 mg/kg;
(I) 0.69 mg/kg/h	(B) 50 IU/kg;
(I) 1,000 IU/h	
Lord et al., 2002	66.6 ± 14.8	No	No	8 weeks	2	32	3/wk, 3.5–4 h	Tinzaparin	4318 IU	(B) 50–75 IU/kg;
(I) NS	
Schrader et al., 1988	54 ± 15.2 (LMWH),
51.6 ± 17.9 (UFH)	No	NS	12 months	8	70 (35/35)	NS, 4.5–5 h	Dalteparin	(B) 34 IU/kg;
(I) 12 IU/kg/h	(B) 62 IU/kg;
(I) 17 IU/kg	
Harenberg et al., 1995	53.4 ± 19.9 (LMWH),
59.1 ± 15.72 (UFH)	NS	NS	NS	NS	20 (10/10)	3–4 h, 4/wk	Dalteparin	(B) 1750 IU
(I) 26.4 IU/kg	(B) 2650 IU;
36.6 IU infusion	
Verzan et al., 2004	NS	NS	NS	NS	NS	66	3/wk, 4–5 h/session	Tinzaparin	40 IU/kg	Mean dose 6262
2300IU/session	
Mahmood et al., 2010	52.1 ± 17.2	NS	NS	NS	NS	20	3/wk, 4–5 h/session	Tinzaparin	(B) 34 IU/kg;
(I) 12 IU/kg/h	(B) 62 IU/kg;
(I) 17 IU/kg	
Gritters et al., 2006	55.2 ± 11.7	NS	NS	NS	NS	8	3/wk, 4–5 h/session	Dalteparin	NS	(B) 62 IU/kg;
(I) 17 IU/kg	
Elisaf et al., 1997	57.1 ± 12.3	NS	NS	NS	NS	36	NS	Tinzaparin	(B) 34 IU/kg;
(I) 12 IU/kg/h	NS	
Notes.

NS Not specified

Risk of bias in included studies

Please refer to Fig. S1 and Table S2 for details of risk of bias assessment. In summary, all included studies in the review were considered to be of poor quality based on risk of bias assessment.

Study outcomes

Primary outcomes

None of the included studies assessed graft or fistula thrombosis. Six studies had reported extracorporeal circuit thrombosis, of which the study by Verzan et al. (2004), though it had mentioned that the LMWH group and UH group were similar with regards to the number of people with this outcome, did not report the exact numbers. Harenberg et al. (1995) reported that LMWH group (tinzaparin) did not differ with UH group as both groups had similar number of events (1/10 each). The remaining 4 studies Borm et al. (1986), Schrader et al. (1988), Saltissi et al. (1999) and Lord et al. (2002), had reported this outcome for the total number of dialysis sessions in each group. The number of extracorporeal circuit thrombosis/number of dialysis sessions encountered for the LMWH group in these 4 studies were respectively 4/10, 80/5045, 17/1111 and 32/378 compared to 4/10, 69/5197, 35/1141, 21/382 in the UH group. The risk ratio comparing LMWH to UH in these 4 studies were 1.00 (0.34–2.93), 1.19 [0.87, 1.64], 0.50 [0.28, 0.89], and 1.54 [0.90, 2.62] respectively (Table 3). The pooled risk ratio was 1.00 with a 95% CI [0.62–1.62]. (Fig. 2).

Figure 2 Forest plots: extracorporeal circuit thrombosis.

Forest Plots comparing LMWH Vs UH for extracorporeal circuit thrombosis.

Table 3 Summary table for meta-analysis (extracorporeal circuit thrombosis).

Table detailing the event rates of comparison between LMWH and UH for extracorporeal circuit thrombosis.

Study	LMWH	UFH	Risk ratio	95% CI	
	Events	No. of HD
sessions	Events	No. of HD
sessions			
Borm et al., 1986	4	10	4	10	1.00	0.34–2.93	
Schrader et al., 1988	80	5,045	69	5,197	1.19	0.87–1.64	
Saltissi et al., 1999	17	1,111	35	1,141	0.50	0.28–0.89	
Lord et al., 2002	32	378	21	382	1.54	0.90–2.62	
Total	133	6,544	129	6,730	1.00	0.62–1.62	

Secondary outcomes

Of the secondary outcomes, deep vein thrombosis and pulmonary embolism were not reported by any of the included studies. Two studies (Lord et al., 2002; Saltissi et al., 1999) had addressed vascular compression times. The vascular compression times for LMWH (Tinzaparin) (9.5 ± 3.0 min) compared to UH (9.5 min ± 1.8 min) were similar in the study by Lord et al. (2002). In Saltissi et al. (1999), vascular compression time for LMWH (enoxaparin) (388 ± 164 s) was similar to that for UH (331 ± 135 s). In total there were 5 studies that had addressed one or more of the lipid profile components. The study by Mahmood et al. (2010) had reported acute changes in triglyceride levels during dialysis with LMWH (Tinzaparin) compared with UH. However, Saltissi et al. (1999), reported lipid changes at 12 weeks and observed that there were no differences in the lipid changes from baseline when LMWH (enoxaparin) was compared to UH for LDL, HDL, total cholesterol, VLDL and triglycerides. However, Schrader et al. (1988) reported that UF group had significantly higher triglyceride and VLDL cholesterol levels compared to UH group (Fragmin) (P < 0.05) at the end of 12 months but the groups were similar with LDL and HDL levels. Gritters et al. (2006) observed that there were no differences between the LMWH group and the UH group with regards to LDL levels measured at 1 week after study initiation. Elisaf et al. (1997) showed that total cholesterol, triglycerides, LDL cholesterol and total cholesterol/HDL ratio significantly decreased after LMWH (Tinzaparin) switch from UH at 3 months, 6 months and 12 months but HDL cholesterol did not significantly change during this period. Four studies had reported bleeding complications (Borm et al., 1986; Schrader et al., 1988; Saltissi et al., 1999; Lord et al., 2002). The number of events/patient in each group were respectively 2/10, 3/32, 12/36, and 19/35 for the LMWH group and 1/10, 8/32, 6/36 and 16/35 for the UFH group. The risk ratio for bleeding in the LMWH group for Borm et al., 1986; Schrader et al., 1988; Saltissi et al., 1999; Lord et al., 2002 compared to UFH were respectively 2.00 (0.21–18.69), 0.38 (0.11–1.29), 2.00 (0.84–4.75) and 1.19 (0.74–1.90) (Table 4). The pooled risk ratio from meta-analysis of these 4 studies was 1.16 (0.62–2.15) (Fig. 3).

Figure 3 Forest plots: bleeding complications.

Forest plots comparing LMWH Vs UH for bleeding complications.

Table 4 Summary table for meta-analysis (bleeding complications).

Table details the event rates comparison between LMWH and UH for bleeding complications.

Study	LMWH	UFH	Risk ratio	95% CI	
	Events	No. of
Patients	Events	No. of
patients			
Borm et al., 1986	2	10	1	10	2.00	0.21–18.69	
Schrader et al., 1988	19	35	16	35	1.19	0.74–1.90	
Saltissi et al., 1999	12	36	6	36	2.00	0.84–4.75	
Lord et al., 2002	3	32	8	32	0.38	0.11–1.29	
Total	36	113	31	113	1.16	0.62–2.15	

Discussion

In this systematic review and meta-analysis we observed that LMWH was similar to UH with regards to extra-corporeal circuit thrombosis and bleeding complications. We did not find studies that assessed graft/fistula thrombosis and prevention of deep vein thrombosis and prevention of pulmonary embolism

To the best of our knowledge only 1 other review by Lim, Cook & Crowther (2004) has been published in this similar topic. The important differences with that review are: (1) they had used LMWH that was approved in Canada and we had used LMWH approved in the United States. Hence, 5 trials that had used Nadroparin (LMWH approved in Canada but not in the US) were not included in our review (Janssen et al., 1996; Reach et al., 2001; Stefoni et al., 2002; Nurmohamed et al., 1991; Liu & Wang, 2002), (2) they had used studies that had citrate and other anti-coagulants as control group whereas; we preferred to use only UH as control. Hence, 3 more trials that they had used were not in our review for this reason (Apsner et al., 2001; Polkinghorne, McMahon & Becker, 2002; Beijering et al., 2003; Anastassiades et al., 1990; Anastassiades et al., 1989), (3) They had used factor Xa levels as one of the outcomes that denoted adequacy of action of LMWH, but with the current available evidence, factor Xa levels have been found not useful in clinical monitoring of LMWH efficacy and hence we opted not to use this outcome for our analysis (Van Veen et al., 2011). However, similar to their review, we found that LMWH was similar to UH with respect to efficacy (extracorporeal circuit thrombosis) and safety (bleeding complications) with LMWH approved for use in the US. However, since their review had larger number of studies included for meta-analysis, their sample size was larger, and hence estimates were more precise than ours. Only the study by Saltissi et al. (1999) showed LMWH to be strongly protective for extracorporeal thrombosis, while our pooled estimate and that by the review by Lim, Cook & Crowther (2004) showed null association. The reason for this observation is unclear. There was no obvious difference between this study and the other studies. It might just mean there was a sampling variability in this study as adequacy of dialysis and severity of uremia was not reported in this study. Also, among the LMWHs, Tinzaparin is preferred in ESRD patients considering its higher molecular weight and lesser dependence on renal functions for elimination from the body. However, due to small number of studies included in the meta-analysis, a subgroup analysis assessing individual LMWHs was not possible. Hence, more RCTs are needed to assess comparative efficacy of one LMWH over the other.

Though we searched three large databases for this review, 5 non-English citations were excluded also, 25 citations were un-retrieved so far. Further, we only included trials that had explicitly mentioned random allocation. It is possible that some trials, though random allocation was not mentioned, may have actually randomized their participants. Hence, due to all these factors it is possible that our final included citations may not be all inclusive. However, since this is possibly random and does not have a systematic component to it, it can tilt the effect estimate more towards null. Hence, more extensive search is needed before we draw conclusions from the study.

Small sample size was an important limitation of our included trials. Not only does it reduce the precision of our estimates, it creates doubts if the randomization was indeed adequate. Some included studies had just 8 to 12 patients, and it is possible that randomization did not work properly and hence some confounding still remained. Further, blinding of outcome determination was unclear and attrition was largely unreported in most trials. Also, there was significant clinical, statistical and methodological heterogeneity in the patient characteristics and the outcome determination. Hence, it is possible that biases like observer bias, could have existed in our results and our pooled analysis. Also, considering the research question it is likely that this proposed observer bias might have been for the LMWH groups as investigators expect more bleeding in this group. This ideally should have shifted the effect estimate more in favor of LMWH. But our results showing null association are reassuring because even in the presence of observer bias favoring LMWH our results show null association.

Since most studies included for the review were of poor quality, better RCTs with larger sample size, better randomization protocol and reporting should be conducted. It is surprising that even though these 3 drugs are approved by the FDA, only one of the studies were conducted in the United States (Naumnik, Borawski & Mysliwiec, 2003). In effect, we are using drugs on American people based on trials conducted elsewhere. Although the other studies did involve patients of European descent, it is essential to retest this RCT with other ethnic groups in the US because FDA approval is not specific to European Americans but for every other ethnicity.

Conclusions

From our review findings and those from Lim et al., we may infer that it may be safe to use the three FDA approved LMWH in ESRD patients, without known hypercoagulable states other than the ESRD that they suffer, receiving regular intermittent hemodialysis.

Supplemental Information

Supplemental Information 1 Search strategy

Search strategy for Medline, Embase and Cochrane library.

Click here for additional data file.

Supplemental Information 2 Study Protocol

Detailed study protocol used to perform the study.

Click here for additional data file.

Table S1 Comorbid conditions of the study participants in the included studies

Comparison of co-morbid conditions between the 2 groups.

Click here for additional data file.

Figure S1 Risk of Bias figure

Figure details the risk of Bias assessment followed in the included studies.

Click here for additional data file.

Supplemental Information 5 PRISMA check list

Details the PRISMA Check list we followed.

Click here for additional data file.

Table S2 Risk of Bias Table

Details the risk of Bias assessment followed in the included studies.

Click here for additional data file.

Data S1 Data file

Contains all retrieved citations.

Click here for additional data file.

Additional Information and Declarations

Competing Interests

Author Contributions

The authors declare there are no competing interests.

Ghanshyam Palamaner Subash Shantha conceived and designed the experiments, performed the experiments, analyzed the data, contributed reagents/materials/analysis tools, wrote the paper, prepared figures and/or tables, reviewed drafts of the paper, developed the search strategy.

Anita Ashok Kumar conceived and designed the experiments, performed the experiments, analyzed the data, contributed reagents/materials/analysis tools, wrote the paper, prepared figures and/or tables, reviewed drafts of the paper.

Mansha Sethi performed the experiments, contributed reagents/materials/analysis tools, wrote the paper, reviewed drafts of the paper.

Rohit C. Khanna conceived and designed the experiments, performed the experiments, analyzed the data, contributed reagents/materials/analysis tools, wrote the paper, reviewed drafts of the paper.

Samir Bipin Pancholy conceived and designed the experiments, performed the experiments, analyzed the data, wrote the paper, prepared figures and/or tables, reviewed drafts of the paper.

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
