# Peer review of "Efficacy and safety of low molecular weight heparin compared to unfractionated heparin for chronic outpatient hemodialysis in end stage renal disease: systematic review and meta-analysis"

_PeerJ, doi:10.7717/peerj.835_

## Round 0.1 · original submission · Major Revisions

As you can see, a number of remarks and suggestions have been made that, in my view, are both valid and useful. I suggest you to take them in full consideration and have a fresh look at your manuscript and see how it could be improved.

Reviewer 1 ·

Basic reporting

The authors carried out a systematic review and meta-analysis of randomized clinical trials to evaluate the efficacy and safety of low molecular weight heparin (LMWH) compared to unfractionated heparin (UH) in end stage renal disease (ESRD) patients receiving outpatient, chronic, intermittent hemodialysis. Based on the reported results, LMWH is as safe and effective as UFH. However, given the poor quality of the studies included, larger trials are warranted before drawing firm conclusions.

Experimental design

Systematic Review and Meta-analysis of Randomized Controlled Trials.

Validity of the findings

Limited validity due to methodological limitations.

Additional comments

Remarks to the Editor
The research focus is appealing and interesting to a research agenda. The use of the methods applied by the Cochrane Collaboration confers strength to the authors’ work.
However, this study suffers from some limitations (reported below) which refrain me providing full support to its publication in PeerJ.

Major compulsory revisions

In both the abstract and methods (criteria for considering studies for this review) English language is mentioned as an applied limitation. Conversely, when describing the search methodology, the text reads “We did not use language or human study restrictions when we searched for citations”. This is confusing to the reader.
However, the number of non English articles judged eligible was not negligible, i.e., 23/216. This, along with the impressive number of manuscripts not retrievable by the deadline reported, i.e., 83, importantly limits the quality of this systematic review. The authors should consider both including non English mother tongue collaborators in their working group and improving their protocol for contacting authors of the manuscripts judged eligible.

The authors should report on whether an expert librarian was involved in the design of the search strategy.

I do not fully understand the need for consensus in the title and abstract screening phase. In my view, this might have additionally biased the results of the systematic review.

I have concerns regarding the lack of consideration of co-interventions, particularly when the features of the study participants are considered, These patients suffer with end stage renal disease mainly from diabetes and hypertension. How can the authors completely exclude the impact of co-interventions on either the primary or secondary outcomes?

·

Basic reporting

- Even if the authors followed the PRISMA statement, the authors should report their work in a continuous way. There is too much sub-sections that further bring confusion. This work should be more narrative for better clarity. The authors should really focus their work on the rational and explained already in the introduction what is really investigated in this study. To date, the manuscript is very difficult to understand and need to be re-written according to the comment raised above.
- Is the meta-analysis recorder in a meta-analysis registry (i.e. METCARDIO, PROSPERO)?

Experimental design

- The choice of a random or fixed effect model should not be based on the degree of heterogeneity but rather on the heterogeneity between study design, outcome, etc. The choice of a random or a fixed effect model should be preferably performed before the assessment of the heterogeneity.
- The authors cannot exclude some populations based on the fact that there are few RCTs in a specific population. In addition, the aim of a meta-analysis is not to preserve homogeneity but to take into account all studies that fulfil the criteria of inclusion (mentioned previously in a dedicated protocol) and discuss on the possible heterogeneity. After, it could be interesting to perform stratifications by subgroups of studies and also to perform sensitivity analyses to assess the robustness of the results.
- The authors included only ESRD patients receiving chronic, intermittent, outpatient haemodialysis. However, this question is never clearly exposed in the introduction of the manuscript and what we expect when reading the title and the introduction is that the authors performed a meta-analysis comparing the efficacy and safety of LMWH versus UFH in ESRD for anticoagulant therapy such as pulmonary embolism, deep vein thrombosis, etc. Therefore, the authors should better describe the aim and the context of the study using standard format for publication of meta-analysis and systematic reviews.

Validity of the findings

- The authors aimed at investigating graft or fistula thrombosis as primary outcome. However, no study included in this work assessed this outcome. I think that the main question is missed and that the authors should reiterate their main question.
- The authors failed to include many of their primary or secondary outcomes in the meta-analysis since there is only few studies that reported the outcome of interest. On the other hand, the authors discussed outcomes not considered of interest for their review. What is the interest to mention them?

Additional comments

- Importantly, the authors should have a reflexion regarding the type of LMWH. It has been showed that tinzaparin is preferred in patients with ESRD due to higher molecular weight and less important contribution of renal function as route of elimination. This should be discussed in this review.
- The methodology is too detailed with useless statements that further bring confusion for the reader. Please follow the PRISMA statement and avoid unnecessary details.
- Please show consistency when using abbreviations (UFH vs. UH; LMWH vs. LMW heparin, etc)
- Comments on the introduction regarding the pharmacological point of view:
- P3 description of the intervention: one of the main advantage of LMWH is a longer half-life that allows a once daily/twice subcutaneous administration. Antithrombin III is now simply called antithrombin. Antithrombin does not lyse the clot, this is the role of plasminogen. Heparin was shown to markedly accelerate the inhibition of thrombin, and also factors (F) Xa, IXa, XIa and XIIa. LMWHs, due to the shorter polysaccharide chain show a more pronounced FXa inhibitors profile. However, it depends on the length of the polysaccharide chain and therefore, all LMWHs do not show the same inhibitory profile. This should be corrected. (See reference 9)
- P4: The authors are not expected to know if there is (or not) enough studies to perform the meta-analyses. Avoid these kinds of statement in your manuscript.
- The grammar could be improved.

---

## Round 0.2 · accepted · Accept

I was favourably impressed by the quality of your rebuttal and revsion, and trust that the new manuscript will be more convincing for the readers than its original version.